# A Method to Predict Water Recovery Rate in the Collection and Froth Zone of Flotation Systems

**Jose Martinez [1], Miguel Maldonado [1,\*] and Leopoldo Gutierrez [2]**

[1]  Departamento de Ingeniería Metalúrgica, Universidad de Santiago de Chile, 8320000 Santiago, Chile; jose.martinez.p@usach.cl

[2]  Departamento de Ingeniería Metalúrgica, Universidad de Concepción, 4030000 Concepción, Chile; lgutierrezb@udec.cl

\*  Correspondence: miguel.maldonado.s@usach.cl

**Abstract:** This paper describes a method to predict water recovery rate into and through the foam in a bubble column operating under different gas rates, froth depths, and frother types and concentrations. Three frothers were considered: Metil Isobutil Carbinol (MIBC), a proprietary blend of alcohols, aldehydes, and esters commercialized under the name PINNACLE® 9891, and a PGE-based Dow Froth 1012 (DF1012). The water rate entering into the froth (foam) layer from the bubbly (collection) zone was estimated as the water rate overflowing the column when operating at a thin stable foam layer, i.e., 0.5 cm. It was observed that the rate at which water entered into the froth phase could be modelled as a unique linear function of the gas holdup below the froth, regardless of the frother chemistry. This is a fundamental result not previously found in the literature that also facilitates the calculation of the froth zone water recovery for deeper froths. The water recovery in the froth was found to be an inverse logarithmic function of the average liquid residence time in the froth. Although the same trend was observed for the three frothers tested, they did not converge into a single function, which suggests that frother chemistry plays a role in determining froth structure and then needs to be incorporated when modeling water transport in the froth. Finally, the water overflow rate calculated as the product of the water rate into the froth and froth water recovery predicted the actual measured values fairly well. The water transport model here proposed provides a simple representation of the interactions between collection and froth zone and its relation to easily measure operating variables.

**Keywords:** water recovery rate; froth recovery; modeling; gas holdup

## 1. Introduction

The amount of water reporting into the concentrate stream has been related to the nonselective recovery of fine liberated gangue particles [1,2]. Therefore, consideration of the current industrial trend of processing massive volumes of lower grade complex ores demands a better understanding of the operating variables that govern the transport of water into and through the froth to aid in the development of operating policies that improve selectivity against nonsulfide gangue [3].

The transport of water in a flotation machine can be regarded as a two-step process: water enters the froth zone from the collection zone and then reports to the concentrate launder from the froth [4]. To better control water recovery in a flotation machine, it would be instrumental to relate the collection and froth zone water recoveries to some local operating process variables. Nevertheless, only a few models have followed this two-step process approach to model water transport. Lynch et al. [5] and more recently, Harris [6] modelled water recovery from the pulp phase to the froth phase as a first-order kinetic process, assuming a fully mixed collection zone. This model is only an empirical approximation, as the mechanism that explains water transport differs from that of hydrophobic particles recovered

by true flotation. The water recovery in the froth zone was modelled as a decreasing exponential function of the average residence time of air bubbles in the froth phase, estimated as the ratio of the froth height to the gas rate [7]. Moys [8] assumed that water entered the froth zone from the collection zone in a film surrounding the ascending bubbles (bubble film theory), which could then be modelled as a function of the bubble surface area flux and an equivalent bubble water film thickness [9]. The recovery of water in the froth was again modelled as a function of the air retention time in the froth.

Other methods have applied an integrated approach focusing on predicting the overall water recovery rate. In this case, we encountered models that relate the concentrate water recovery rate to the solid recovery rate [10–13]. Although useful for analysis and design, this model does not provide insight into how to modify the water recovery rate using commonly used manipulated variables. Other models based on the transport of water in the froth phase have also been reported [14–16]. The work of Neetling et al. [14] stands out, as it provides an equation derived from fundamental principles that relate water overflow recovery rate (concentrate flowrate) to the prevailing operating conditions, i.e., pulp properties (density and viscosity), gas dispersion property (superficial gas velocity), and froth properties, i.e., top of the froth average bubble diameter and air recovery. Although that relationship provides some guidance on how to control water recovery in flotation systems, it involves measurements that are challenging to perform in practice, such as averaging the bubble size of the overflowing froth and air recovery, and is prone to error propagation [13,14,17].

This paper aims to contribute to the understanding of the process of water transport in both the collection and froth zones of flotation systems and its relationship to operating variables.

## 2. Experimental

### 2.1. Column Set Up

A lab flotation column (1) of 4 inches' diameter and 2.5 m height made out of transparent PVC, as illustrated in Figure 1, was used to conduct the experiments. A top section made of acrylic (2) was implemented to facilitate froth–liquid interface detection. Air was dispersed into bubbles through a porous tube metallic sparger having an average porous size of 5 μm, which was installed horizontally at the bottom of the column (3). The air flow rate was monitored and controlled using a flow controller (4) (MKS model MFC GE30) connected to a computer using Ethernet. Gas holdup below the froth was calculated from the difference in hydrostatic pressure at two points separated by 450 mm using a differential pressure transmitter (5) (ABB model 266DSH). The pressure signal was recorded using a data logger (model EL-USB-4) with a sampling time of 1 s. The water feed flowrate was measured using a 0.5 inch magnetic flowmeter (6) (SIEMENS model 911/E-IP68 SITRANS FM Transmag 2). A frother solution with a target concentration was prepared using tap water, and was conditioned in the reservoir tank (7) by recirculating the solution using a centrifugal pump (8) with valve A closed and valve B open for around 10 min. The flow of solution to the column was regulated within 2 ± 0.1 L/min by manipulating valves A and B. Froth height was regulated using the column discharge valve C. The water that overflowed the column reported first to the launder and then back into the reservoir tank. The water overflow rate was calculated from the measurement of the volume of water collected in a graduated cylinder beaker per unit of time, and the column of the cross-sectional area. This procedure was carried out four times for each test, and the calculated water recovery rates averaged to halve the standard deviation of individual measurements. For every change in frother concentration, the column was emptied and the water was collected in the reservoir tank where the new solution was conditioned for at least five minutes before being fed back to the column. The experiments were carried out at an ambient temperature of approximately 20 °C.

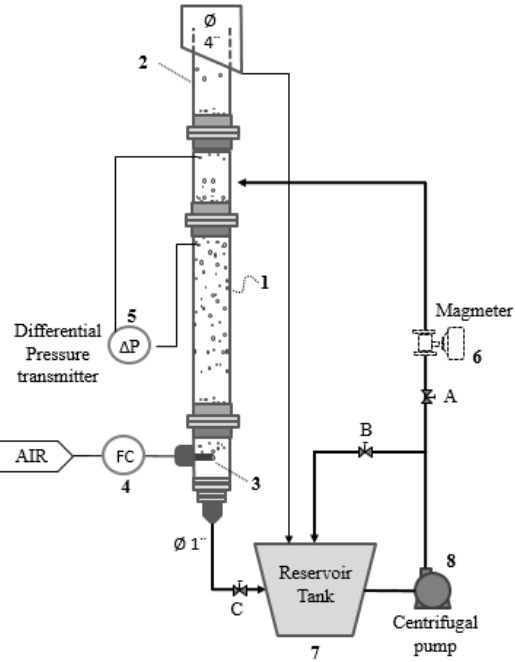

**Figure 1.** Schematic diagram of the apparatus. FC: flow controller.

*2.2. Test Conditions*

Three types of frothers were used in this investigation: MIBC (alcohol, provided by Sigma Aldrich), PINNACLE® 9891 (a proprietary blend of alcohols, aldehydes, and esters, provided by Nalco), and DF1012 (high molecular PGE-based frother, supplied by Dow Chemicals). Table 1 summarizes the conditions tested in the flotation column. For the three types of frothers considered, changes in gas rate, froth depth, and frother concentration were established, and the water overflow rate was measured once a steady condition was reached. The gas rate was varied in a middle range, i.e., from 0.8 cm to 1.6 cm/s. This was enough to produce a stable foam while also allowing the foam–water interface to be accurately identified. The frother concentration range was from moderate to high. Concentrations lower than 10 ppm were avoided, as they produced thin layers of foam requiring high gas rates, which, in turn, induced errors in the froth depth measurements.

**Table 1.** Tested conditions in the flotation column.

| Frother Type | Molecular Weight (g/mol) | Superficial Gas Velocity (cm/s) | Frother Concentration (ppm) | Froth Depth Range (cm) |
|---|---|---|---|---|
| MIBC | 102.17 | 0.8–1–1.2–1.4–1.6 | 10–15–20–50 | 0.5–13.5 |
| PINNACLE® 9891 | not available | 0.8–1–1.2–1.4–1.6 | 10–15–20–50 | 0.5–27 |
| DF1012 | 397.95 | 0.8–1–1.2–1.4–1.6 | 10–15–20–50 | 0.5–26 |

## 3. Results

*3.1. Collection Zone Water Carrying Rate*

The water rate into the froth, here represented by its superficial velocity denoted as JwI, was estimated as the water overflowing the column when operating at a stable thin foam layer, i.e., 0.5 cm. Figure 2 shows the impact of frother type, concentration, and gas rate on the transported water into the froth. It can be observed that water rate into the froth increases with frother concentration and gas rate. In general, for a given gas rate, water entrained into the froth increases with frother concentration, probably due to a reduction of bubble size and an increase in the thickness of the bounded water film [18]. It can also be observed that solutions of DF1012 tend to introduce more liquid



into the froth than those observed for MIBC and PINNACLE 9891. This was expected, as DF1012 is a strong high molecular weight frother.

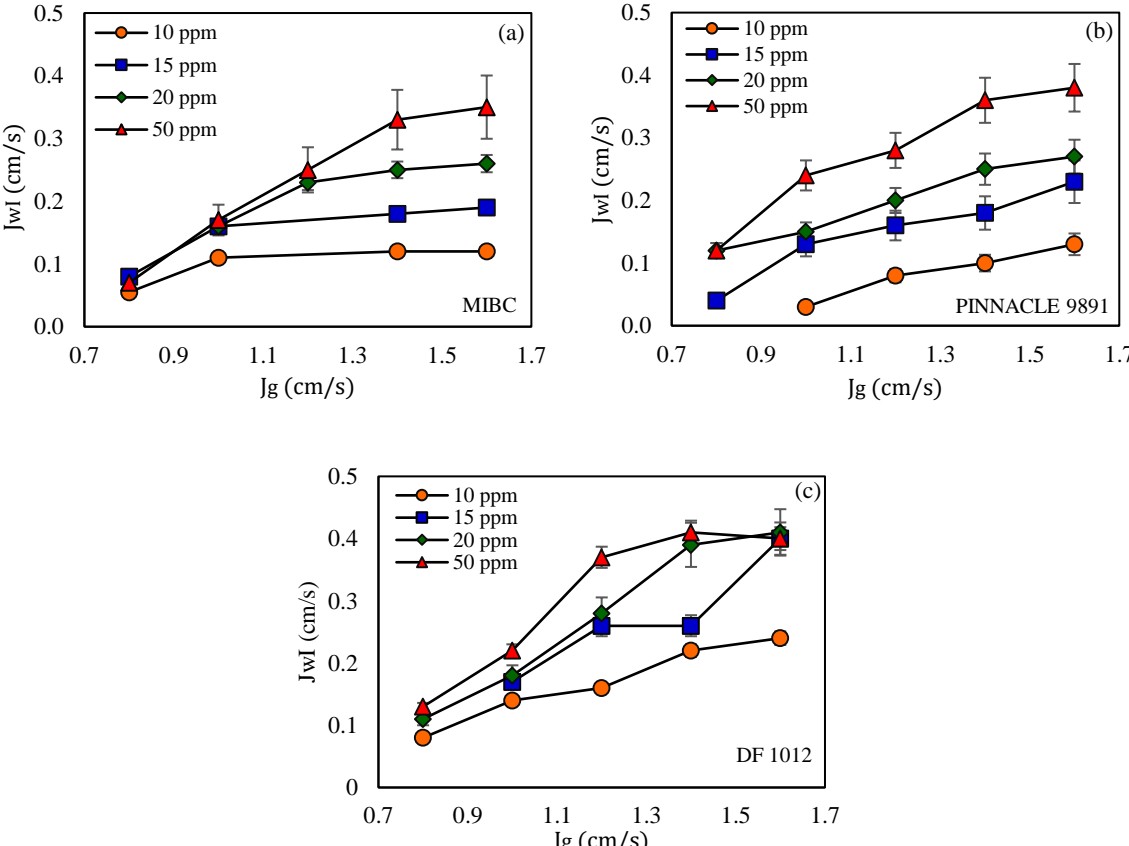

**Figure 2.** Water rate into the froth as a function of gas rate and frother concentration for (**a**) MIBC, (**b**) PINNACLE 9891 and (**c**) DF 1012.

It is well known that gas holdup captures changes of bubble size, frother concentration, rise velocity and gas rate; therefore, the relationship between water rate into the froth and gas holdup was tested for the three frother types, as shown in Figure 3. An underlying single linear relationship for the three tested frothers, regardless of their chemistry, was observed. Thus, the rate of water entrained into the froth from the bubbly zone can be predicted from gas holdup measurements below the froth ($\varepsilon_g$) as follows:

$$\hat{J}_{wI} = 2.7112 \cdot \left( \varepsilon_g - 7.0559\% \right) \tag{1}$$

where a minimum gas holdup of around 7% was required to produce a nonzero water overflow. The symbol ˆ stands for estimate or prediction.

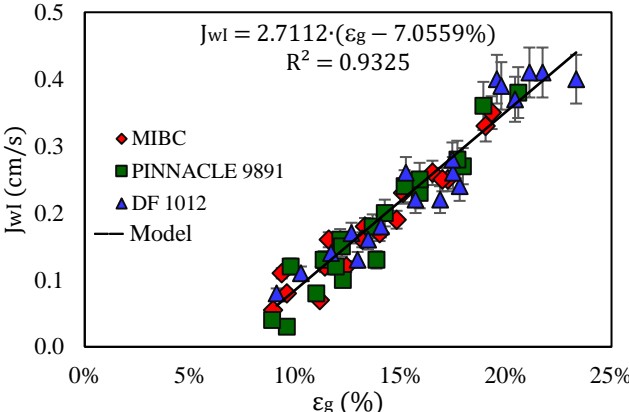

**Figure 3.** Water rate into the froth as a function of gas holdup for the three frothers tested.

### 3.2. Froth Zone Water Rate

Figure 4 shows the dependence of the water overflow rate ($J_{wO}$) on froth depth ($H_F$) for different frother types and concentrations. For the sake of brevity, only results for the maximum superficial gas veleocity tested, i.e., $J_g$ = 1.6 cm/s, are shown, although similar results were observed for the other experimental gas rates. The water overflow rate decreases with froth depth, as expected. It can also be noted that for a given froth depth, an increase in frother concentration increases the rate of water overflowing the column. Again, for similar operating conditions, solutions of DF 1012 tend to transport more water through the froth.

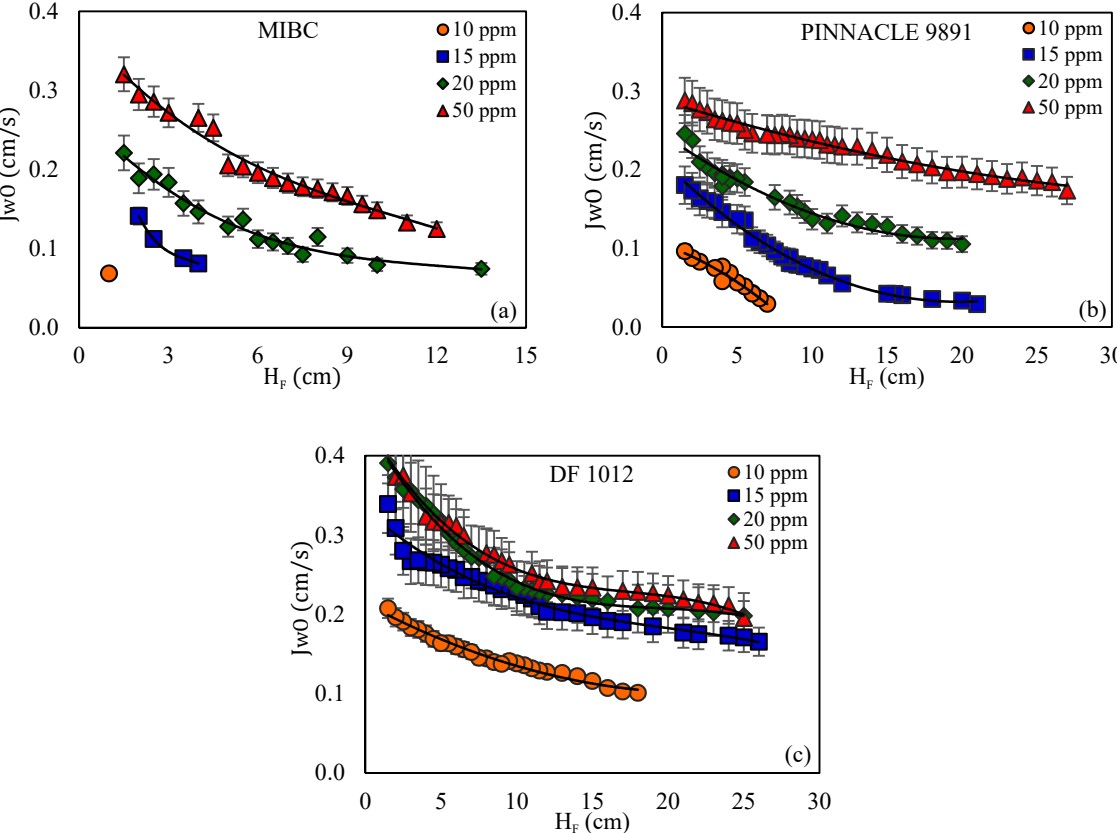

**Figure 4.** Water overflow rate as a function of froth depth for: (**a**) MIBC, (**b**) PINNACLE 9891 and (**c**) DF 1012 and Jg = 1.6 cm/s.

The water recovery fraction in the froth phase can be calculated as the ratio of the measured water rate overflowing the column to the water rate entering the froth estimated from Equation (1):

$$\hat{R}_{wF} = \frac{J_{wO}}{\hat{J}_{wI}} \tag{2}$$

Water recovery in foams has been reported to depend on several factors such as gas rate, froth depth, and froth stability, among others. It has been modelled as an exponential function of the average residence time of air bubbles in the froth ($\tau_{gF}$), defined as the ratio of the volume of air in the froth to the gas velocity leaving the cell as unburst bubbles ($J_{gO}$).

$$\tau_{gF} = \frac{\varepsilon_{gF} \cdot H_F}{J_{gO}} = \frac{\varepsilon_{gF} \cdot H_F}{\alpha \cdot J_g} \tag{3}$$

where $\varepsilon_{gF}$ is the volumetric fraction of air in the froth, $H_F$ is the froth height, $\alpha$ is the air recovery and $J_g$ is the superficial gas rate entering the flotation machine. For practical convenience, Equation (3) is usually simplified to [7]:

$$\tau_{gF} \approx \frac{H_F}{J_g} \tag{4}$$

Figure 5 shows the relationship of the water recovery in the froth phase and the air retention time (using Equation (4)) for all the conditions summarized in Table 1. Although an inverse relationship seems to hold, it is obscured by significant data scattering.

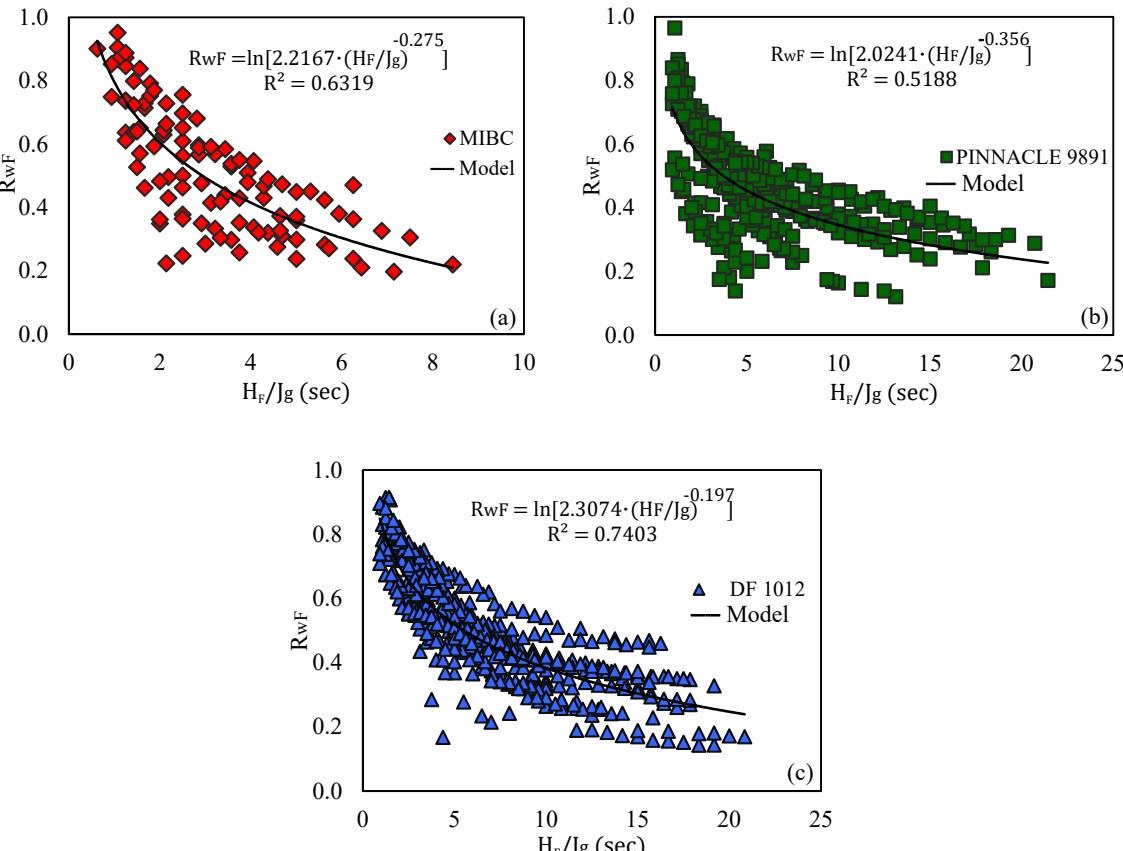

**Figure 5.** Froth water recovery as a function of the average air retention time in the froth ($H_F/J_g$) for (**a**) MIBC, (**b**) PINNACLE 9891, (**c**) DF 1012.

In another attempt to find a better model for the water recovery in the froth, and considering we already have an estimation of the water rate entering the froth, we defined a variable proportional to the average liquid retention time in the froth ($\tau_{wF}$) in analogy to Equation (4) for air.

$$\tau_{wF} \approx \frac{H_F}{\hat{J}_{wI}} \tag{5}$$

Figure 6 shows the relationship between the water recovery in the froth and the average residence time of the liquid in the froth estimated from Equation (5). This variable seems to better predict the recovery of water in the froth at least for MIBC and PINNACLE 9891. A reduction in the coefficient of determination was obtained for DF1012, although the scattering seems similar. A logarithmic function was fitted to the experimental data.

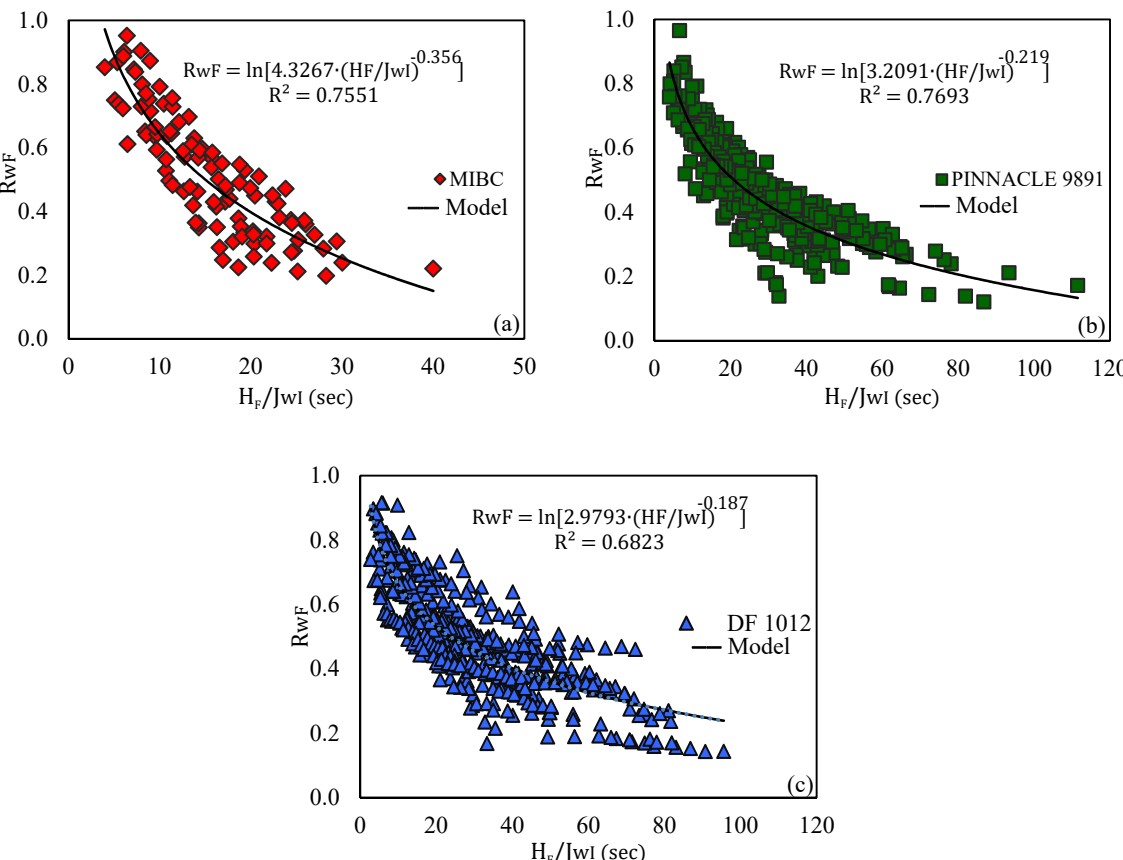

**Figure 6.** Froth water recovery as a function of the average liquid retention time in the froth ($H_F/J_{wI}$) for (**a**) MIBC, (**b**) PINNACLE 9891, (**c**) DF 1012.

Note that if we now replace the water rate into the froth (JwI) with the water rate leaving the cell (JwO) to define the liquid retention in the froth in analogy to Equation (3) for the gas phase, we have:

$$\tau_{wF} \approx \frac{H_F}{J_{wO}} \tag{6}$$

Figure 7 shows that the water recovery in the froth could be reasonably well modelled as a logarithmic function of the liquid retention time in the froth approximated in Equation (6).

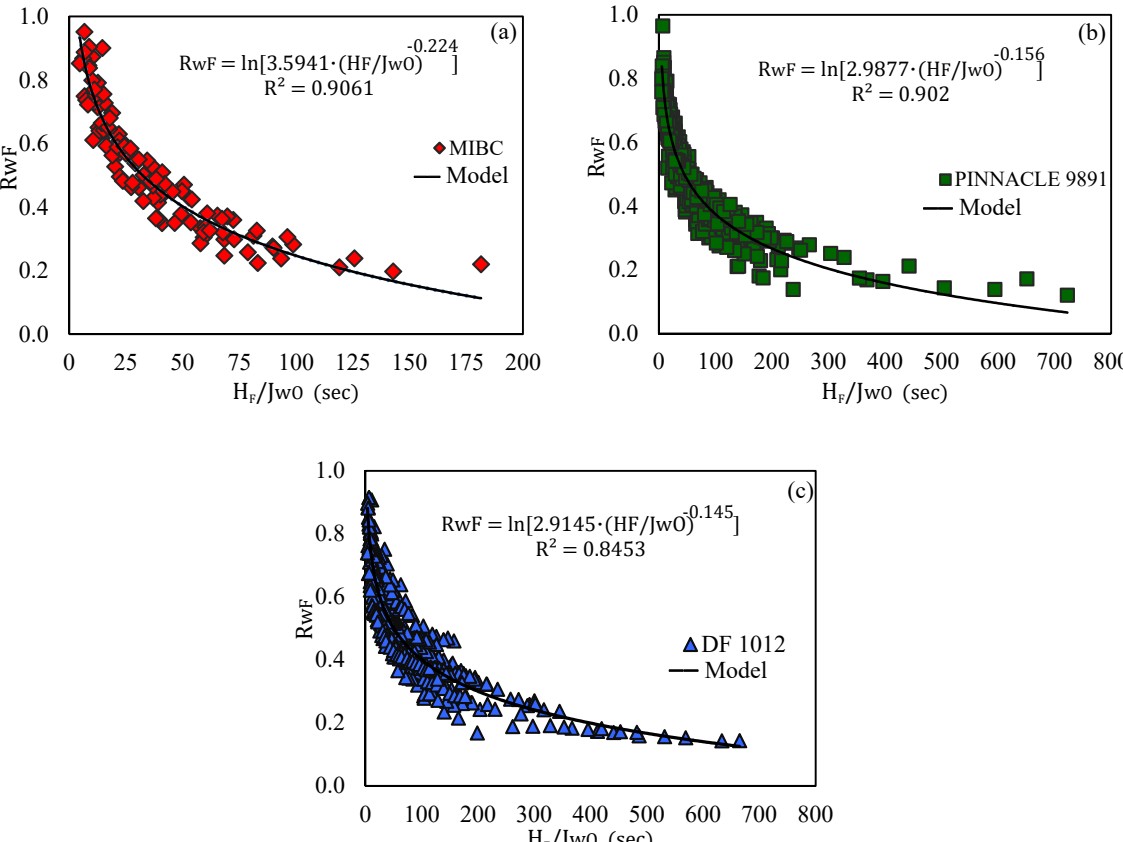

**Figure 7.** Froth water recovery as a function of the average liquid retention time in the froth ($H_F/J_{wO}$) for (**a**) MIBC, (**b**) PINNACLE 9891, (**c**) DF 1012.

Figure 8 shows the fitted log models for the three frothers. DF1012 and PINNACLE 9891 exhibit a similar trend and achieve higher froth water recoveries compared to MIBC.

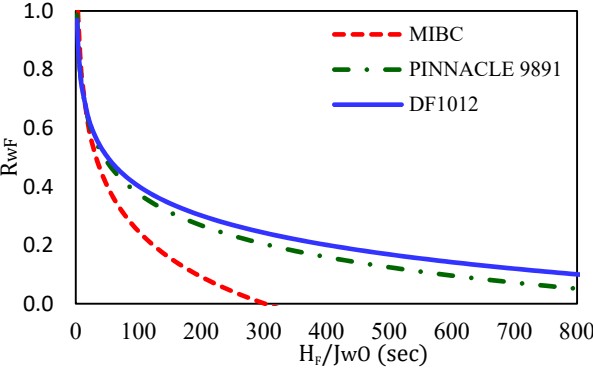

**Figure 8.** Fitted models for the recovery of water as a function of liquid retention time in the froth for different frothers.

Once the recovery of water in the froth is determined, the water overflowing the column can be predicted from the following Equation:

$$\hat{J}_{wO} = \hat{R}_{wF} \cdot \hat{J}_{wI} \tag{7}$$

Note that as the $\hat{R}_{wF}$ depends on $\hat{J}_{wO}$, a nonlinear algebraic equation arises for the water recovery in the froth that must be solved for any value of $\left( H_F / \hat{J}_{wI} \right)$, i.e.,:

$$\hat{R}_{wF} = \ln\left[ a \cdot \left( \frac{H_F}{J_{wO}} \right)^b \right] = \ln\left[ a \cdot \left( \frac{H_F}{\hat{R}_{wF} \cdot \hat{J}_{wI}} \right)^b \right] \tag{8}$$

where a and b are parameters that depend on frother chemistry. As expected, the solution of algebraic Equation (8) will provide similar results as the models shown in Figure 6, since the same input variables were considered, namely $H_F$ and $\hat{J}_{wI}$. Figure 9 shows a block diagram representing the model. It provides a simple representation of the interactions between collection and froth zone and their relationships to easily measure the operating variables.

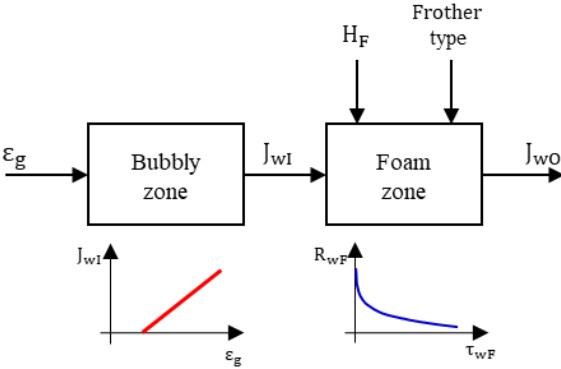

**Figure 9.** Block representation of the mathematical model to predict water recovery rate.

Finally, Figure 10 shows the predicted versus the measured water overflow rate following the procedure illustrated in Figure 9. Despite some data scattering, the model can explain reasonably well the main variations in the experimental data.

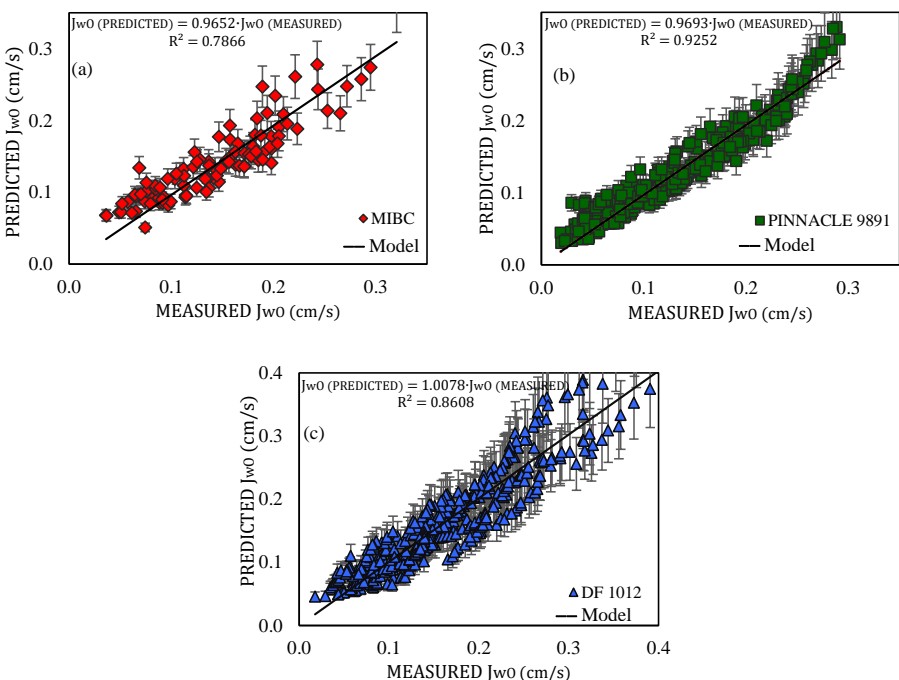

**Figure 10.** Predicted versus measured water overflow rate for (**a**) MIBC, (**b**) PINNACLE 9891 and (**c**) DF 1012.

## 4. Discussion

Bubbles transport water in a thin film, a wake, and also by the mechanical action of ascending swarms [4]. It has been found that neither of the first two proposed mechanisms accounts for the amount of water observed in the overflow of flotation systems. Finch et al. [18] reported the thickness of the bound water layer for four frother chemistries ranked from weak to strong (n-Pentanol, MIBC, DF250, and F150). They found that the thickness of the water film was related to the frother chemistry with measured thicknesses ranging from less than 160 nm for n-Pentanol up to 1100 nm for the strongest one, F150. Although the measured water thickness correlated with the measured water overflow rate, this alone could not explain the amount of water observed in the experiments [19]. In this study, a unique linear relationship between the water rate into the froth and gas holdup below the froth was observed. This relationship proved to hold for different frother types, concentrations, and gas rates and, therefore, lends some support for the bubble swarm theory as the primary mechanism for water transport into the froth. Note that an increase in the thickness of the bounded water surrounding the bubbles will also be detected by gas holdup as bubble rise velocity will be reduced.

Moyo et al. [20] studied the dependence of the water overflow rate on gas holdup in a bubble column as a means to characterize frother types. For a given frother chemistry and constant froth depth (7 cm), a unique linear $J_{wO}–\varepsilon_g$ relationship was observed, even when spargers having different porosities were used. Although the linearity was preserved for different frother types, not a single function between $J_{wO}$ and $\varepsilon_g$ was found; instead, parallel straight lines having different gas holdup axis intercepts were observed. These different intercepts corresponded to the minimum gas holdup required to produce a nonzero water overflow rate, and varied from weak to strong frothers, a property exploited for frother characterization. Higher froth depths were also tried, e.g., 15 and 35 cm, but for these cases, $J_{wO}–\varepsilon_g$ straight lines were no longer parallel [19]. The results obtained in this study for a thin froth depth, where a unique $J_{wO}–\varepsilon_g$ relationship was observed for different frother types, suggests that it is the effect of the frother chemistry in the froth structure that dictates its classification.

Water transport in the froth was quantified by the water fraction recovery calculated as the ratio of the measured water overflow rate to the estimated water rate entering the froth from gas holdup measurements. It was found that for varied conditions, water recovery in the froth can be described as a function of the ratio $H_F/J_{wO}$, here considered to be proportional to the liquid retention time in the froth. Note that this approximation misses the liquid holdup in the froth, which has been found to depend on the prevailing operating conditions [14,16]. Interestingly, the relationship for water recovery in the froth was not unique, but was a function of the frother chemistry. For example, it revealed similar behavior for DF1012 and PINNACLE 9198, which achieved higher recoveries compared to MIBC. This may well provide another method to characterize frothers, but also suggests that frother chemistry must be incorporated to model water transport through the froth.

The water transport model here proposed provides a simple representation of the interactions between collection and froth zone, making it possible to easily measure operating variables, namely, gas holdup and froth depth. Note that an industrial gas holdup sensor was recently developed [21,22]. Although these results cannot be directly extrapolated to a three-phase system, as the presence of particles can drastically modify froth stability, they suggest a new method to approach the problem.

## 5. Conclusions

This paper proposed a methodology to estimate water recovery rates into and through froth in a column operating in an air-water system. Three frother types were tested: MIBC, PINNACLE® 9891, and DF1012. The dependence of water rate into the froth on gas holdup was found to converge to a single underlying linear function, regardless of the frother chemistry. Water recovery in the froth was found to decrease monotonically with the average liquid retention time in the froth, and to be a function of the frother type. The water overflow rate predicted as the product of the water into the froth and its recovery in the froth can explain reasonably accurately most of the observed variations in the experimental data.

**Author Contributions:** J.M.: data curation, methodology, Conducting experiments; M.M.: methodology, formal analysis, writing—original draft preparation, project administration; L.G.: formal analysis; writing—review and editing. All authors have read and agreed to the published version of the manuscript.

**Funding:** This research received no external funding.

**Acknowledgments:** Gutierrez acknowledges support from Center CRHIAM ANID/FONDAP/15130015.

**Conflicts of Interest:** The authors declare no conflict of interest.

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
