# Peer review of "A Method to Predict Water Recovery Rate in the Collection and Froth Zone of Flotation Systems"

_minerals, doi:10.3390/min10070630_

Round 1

Reviewer 1 Report

REVIEW REPORT

A Method to Predict Water Recovery Rate in the 2 Collection and Froth Zone of Flotation Systems

Line 21: The amount of water “reported”. Should be “reporting”.

Line 32: demands “for” a better. Remove “for”.

Line 36: and then water “reported” to the concentrate. Should be “reporting”.

Line 38: “modeled” should be “modelled”.

Line 56: Although that relationship provides some guidance on how to control water recovery in flotation systems, it involves measurements that are challenging to perform in plant practice and are prone to error propagation (Wang et al., 2015). Specify the measurements you are referring to and why they are difficult to measure.

Line 59: Replace “communication” with “paper”.

Line 60: “The paper is organized as follows: Section 2 describes the experimental setup and test conditions, section 3 presents the impact of frother type and concentration, gas rate and froth depth on the water rate into and through the froth”. “Finally, section 5 and 6 provide some discussion and 64 the main conclusions derived from this work, respectively”. These 2 sentences do not add value to the introduction and should therefore be removed.

The introduction does not explicitly spell out the gap in literature or knowledge that the authors intend to cover. This has to be the unique selling point of the paper. To me, the authors are just replicating the work that was reported by others. The authors say: “A mathematical model that separates the collection and froth zone is presented, and its predictive capabilities tested”. What will make authors’ model different from the models that were predicted by other research. This has to be crystal clear at this point in the manuscript.

Line 68: “Figure 1 shows a schematic diagram of the flotation column setup implemented”. This sentence does not make sense. Please paraphrase it.

Line 69: Replace “comprises a flotation..” with “comprises of a flotation…”

Line 69: Replace “diameter for 2.5 m height” with “diameter and 2.5 m height”

Line 71: What was the porosity of the sparger used in this study.

Line 78: The authors use tap water. In real plant operations, process water containing some ions is used. The ionic strength of water obviously affects both the pulp and froth phase. Wouldn’t the study have been more relevant if synthetic plant water containing some ions was used?

Section 2.1: The authors state that the column material was PVC. Why PVC? Is the wall hydrophobic or hydrophilic? Any comment on wall effects? Any comments on the effect of column diameter and its relationship with wall effects at the column diameter of 4 inches (100 mm)?

Section 2.2: What were the purities of the 3 frothers investigated? How were the purities determined? Were the effect of impurities taken into account when interpreting the results obtained in this study? What was the distribution of alcohols, aldehydes, and esters in PINNACLE 9891 and what was the average molecular weight of the frother?

Table 1: The authors must justify the choice / selection of the operating conditions, i.e. frother concentration, superficial gas velocity and froth depth. What are the critical coalescence concentrations (CCC) of the different frothers investigated in this study? Were the CCCs taken into account when selecting the range of frother concentrations to be investigated in this study.

Figure 2: Earlier on, the authors mention that: “This procedure was carried out four times for each test, and the calculated water recovery rates averaged to halve the 85 standard deviation of individual measurements”. There are no error bars on the graphs to help decide whether the differences were statistically different or not? Please insert error bars.

Line 107: The authors assert that: “For similar conditions, solutions 107 of DF1012 introduced more liquid into the froth than that observed for MIBC and PINNACLE 9891”. I do not entirely agree to this statement. The statement is only true at all frother concentrations, expect at 50 ppm.

Figure 2: More could have been said about these results. A rich comparative study of the 3 frothers is lacking. For example, what is the relationship between Jwl and frother concentration for a given Jg?

Line 118: For the sake of “space”. Replace “space” with “brevity”.

Line 122: Replace “trough” with ‘through”.

Line 133: Replace “in the froth” with “in the froth phase”.

Section 4: The discussion section is very shallow leaving pertinent issues that were highlighted in the results section. In other words, there is no link between the results and discussion of the same. The reviewer feel that the fundamental aspects of frother chemistry have been completely overlooked in a hurried attempt to establish quantitative relationships. For example, reasons for the following results are not discussed:

  1. “It can be observed that water 106 rate into the froth increases with frother concentration and gas rate. For similar conditions, solutions 107 of DF1012 introduced more liquid into the froth than that observed for MIBC and PINNACLE 9891”. What could be the reason(s) for these observed results? This is especially important as it zero in on frother chemistry.
  2. “Three frother types were tested, namely, MIBC, 198 PINNACLE® 9891, and DF1012. The dependence of water rate into the froth on gas holdup was found 199 to converge to a single underlying linear function regardless of the frother chemistry”. The authors must explain this phenomenon in the discussion section. Is it because of the inter-relationships between frother concentration, bubble size and gas rate which all contribute to gas hold up.
  3. In real flotation systems, 3 phases are present, viz: water, air and solids. What could be the effect of solids on the results obtained in this study?

Reviewer 2 Report

Dear Authors,

The aim of the paper is to provide a methodology for prediction of water recovery rate in the collection and froth zone during column flotation and to better understand water transport phenomenon under conditions of using various frothers, and hence, the paper can be of interest of both academia and industry.

Please find below recommendations for improving the paper.

References, lines 219-220, 227-228: The year of publishing was missed.

Moyo, P., Gomez, C.O., Finch, J.A. Characterizing frothers using water carrying rate. Canadian 219 Metallurgical Quarterly, 46:3, 215-220.

There are some slight typing errors in the text:  line 25 ‘froter; line 96 “…Gas rate was varied in a middle range, i.e., from 0.8 cm to 1.6 cm…”; line 122 “…transport more water trough the froth…”.

It will make a paper stronger if the authors can provide information if the model includes the effects of rinse water flow rate or comment on that issue.

Lines 60–65: “…The paper is organized as follows: Section 2 describes the experimental setup and test conditions, section 3 presents the impact of frother type and concentration, gas rate and froth depth on the water rate into and through the froth. A mathematical model that separates the collection and froth zone is presented, and its predictive capabilities tested. Finally, section 5 and 6 provide some discussion and the main conclusions derived from this work, respectively.”

I am afraid the paragraph needs correction as there are no sections 5 and 6 in the paper and section 4 is missed.

The paper will benefit if the authors check dimensions of X and Y axis in Fig. 2 and relate them to JwI and JG in the text. Maybe “specific water flow rate” and “superficial gas velocity” instead of “water rate” and “gas rate”?

“Figure 1. Experimental column setup: (a) illustration and (b) actual photograph.”

I am afraid there is no any photography. The title of Fig. 1 needs correction.

Line 118: “…For the sake of space, only results for Jg = 1.6 cm/s are shown…”

The paper will benefit if the authors support their choice of Jg = 1.6 cm/s from research or industrial point of view.

Line 119: “…although similar results were observed for the other gas rates tested…”

Maybe “similar trends”?

Lines 121–122: “ Again, for similar operating conditions, solutions of DF 1012 tend to transport more water trough the froth…”

There is no any statement about effect of DF1012 in “Results section” before this sentence. Maybe something was missed in previous paragraphs?

In line 140 authors introduce “…residence time of the liquid in the froth estimated…” – τwF, however, in Fig. 6 X-axis is named as HF/JwI. The similar issues arise in Fig. 7 and 8 with τwF. Readers will appreciate if the authors correct these issues.

Lines 154–158: The paper can attract more readers and citations if the authors expand their comments on Eq. (8) and explain how one can use this equation for practical usefulness. Additional comments of the authors on graphical model will be also appreciated.

Joining “Results” and “Discussion” sections may be beneficial as it provides more space for explaining the results. The paper will benefit if the authors include in “Results and Discussion” section guidelines describing a proposed methodology as it mentioned in the conclusions.

In general, the paper presents interesting results which can be considered for publication after slight correction.

Regards,

Reviewer

Reviewer 3 Report

This is an interesting study involving a method to predict water recovery rate into and through the foam in a bubble column operating under different conditions of gas rate, froth depth, and frother type and concentration.

Abstract Needs more information about conclusions of the study: Examples: The dependence of water rate into the froth on gas holdup was found to converge to a single underlying linear function regardless of the frother chemistry. Water recovery in the froth was found to decrease monotonically with the average liquid retention time in the froth and be a function of the frother type. The water recovery in the froth could be reasonably well modeled as a logarithmic function of the liquid retention time in the froth The current abstract is not eye-catching.

Figure 2. Water rate into the froth as a function of gas rate and frother concentration for (a) MIBC, (b) PINNACLE 9891 and (c) DF 1012 : Error bars must be provided, so that the repeatability of the results are known.

124 line : Equation (1) needs a reference

The publication of this paper would help the potential readers to know more about the industrial applications of the theoretical flotation hydrodynamic model. It can be accepted for publication in this nice journal in the area of mineral processing.

Round 2

Reviewer 1 Report

The items raise by the reviewer have been adequately addressed by the authors.